# Global Financial Investment Connectedness, ICT, and Intellectual Property Strategies: A Country-Level Empirical Analysis

**Yonghee Kim** [1] and **Sungjin Yoo** [2,*]

1 OpenRoute Research, Yongin 16953, Republic of Korea
2 School of Business Administration, Soongsil University, Seoul 06978, Republic of Korea
* Correspondence: sjyoo@ssu.ac.kr

**Abstract:** This study investigated the direct and indirect impacts of financial investment connectedness and Information and Communication Technology (ICT) on countries' intellectual property (IP) strategies. By utilizing the panel logit model on longitudinal country-level data, we found that countries' positions in the global financial investment network significantly affect their IP strategies. Furthermore, ICT usage weakens the IP strategies' reliance on global financial investment connectedness. This study is among the first to link financial investment connectedness and ICT to intellectual property strategy. The implications for governments managing financial investment portfolios and making intellectual property strategies are derived.

**Keywords:** financial investment connectedness; ICT; intellectual property strategy; social network analysis; country-level data

## 1. Introduction

Intellectual property (IP) refers to exclusive rights to all creations of the inventions used in commerce, such as patents, copyrights, trademarks, and trade secrets (https://www.wipo.int/about-ip/en/, accessed on 5 March 2024). IP has been considered a powerful business tool because it assures ownership over human intellect and exclusively reserves the right to control the use of a piece of IP. Using IP, such as selling and licensing it, helps organizations and nations earn revenue and profit, thereby securing a competitive position in the market. There are many successful cases of IP strategies used by corporate firms and countries. Among them, the stories of Apple's iPhone and iPad provide examples of the benefits of IP strategies (https://www.ktmine.com/exploring-intellectual-property-at-apple-a-study-of-strategy-and-patterns/, accessed on 5 March 2024). Apple gains profits not only from selling devices such as the iPhone, iPad, and Mac, but also from licensing its IPs (https://www.taxpolicycenter.org/taxvox/who-should-get-tax-revenue-apples-intellectual-property, accessed on 5 March 2024). On the other hand, developed countries or second-mover firms actively purchase IPs from developed or first-mover firms so that they can advance their knowledge and absorb cutting-edge technologies. Thus, an IP strategy, which is a plan for developing, acquiring, managing, and monetizing a portfolio of IP assets, is an essential instrument for firms and nations, and the best strategy (e.g., licensing or purchasing IPs) should be different for each (e.g., Palfrey, 2012 [1]; Reitzig, 2007 [2]).

A growing body of research has examined the impact of IP protection on institutional performance at different levels, such as organizational, regional, and national levels (e.g., Gould & Gruben, 1996 [3]; Reitzig, 2007 [2]; Wen et al., 2013 [4]; Wen et al., 2016 [5]; Zhao, 2006 [6]). These prior studies have mainly focused on the effects and consequences of protecting IPs. Some studies (e.g., Branstetter et al., 2006 [7]) have found that stronger IP protection is beneficial, as it promotes more technology transfers, while other recent

research has asserted that IP protection discourages the innovative activities of developers. More importantly, few studies have yet to investigate the antecedents of IP strategies. What factors shape a firm or country's IP strategy have not been determined. Specifically, the strategies used to balance the tradeoff between developing and purchasing IPs are also understudied. The examples above show that the best IP strategy must vary based on different environments, IP capabilities, etc. IP capability refers to the value capture abilities associated with an IP, particularly in the management of intellectual capital (IC) through IPR in the context of technological innovation (e.g., Reitzig, 2009 [8]). To this end, this research studies the factors determining a country's choice of IP strategies regarding the tradeoff between developing and buying IPs. We especially label a country's IP strategy as proactive when its receipt is higher than its payment for IP usage. On the contrary, a country's IP strategy is denoted as a dependence strategy if its receipt is lower than its payment for IP.

Both countries' internal and external factors should be considered when conducting such an IP strategy. From the external perspective, financial investment connectedness determines the cash flow among countries and embeds implicit knowledge in the network. Prior research has proved the causal relationship between financial investments and innovation outcomes (Kim et al., 2016 [9]). Therefore, we try to link the connectedness of global financial investment to national IP strategies. The global financial investment network is a complex system that connects countries through financial transactions (e.g., Ahn et al., 2022 [10]). In our research context, the global financial network refers to a network of countries' global investments, showing which countries invest in a particular country. Financial investment connectedness refers to the degree of interconnectedness between countries within the global financial network. Previous research about global financial investments has visualized this network (Zhang et al., 2016) [11] and examined the effect of countries' network positions on their stock market movement (Chuluun, 2017) [12] and economic growth (Møller & Rangvid, 2018) [13] and studied how the financial market in Asia is interconnected and how these connections have changed over time (Chowdhury et al., 2019) [14]. From an internal perspective, a country's adoption level of Information and Communication Technology (ICT) significantly determines its citizens' access to information. It provides opportunities for them to exploit this information in many sectors (e.g., healthcare, traffic conditions, logistics, and so on). Earlier studies have proven that ICT usage affects countries' economic productivity (Dedrick et al., 2013) [15], political corruption (Srivastava et al., 2016) [16], and social well-being (Ganju et al., 2016) [17]. Unlike prior studies, we examine the impact of ICT usage on a country's IP strategy choice.

Therefore, this paper will investigate financial investment connectedness's direct and indirect impacts on countries' IP strategies. In addition, we will study its dependence on the country's financial investment network, which varies based on the country's ICT usage level. By utilizing the panel logit model and longitudinal country-level data from 2001 to 2015, we found that countries' global financial investment network positions significantly affect their IP strategies. Furthermore, from the perspective of sustainability in technology management, countries' ICT usage weakens their IP strategy's reliance on global financial investment connectedness, thereby saving resources and promoting a sustainable situation for some developing countries. This study is among the first to link financial investment connectedness and ICT to IP strategies.

## 2. Hypotheses Development

Nations can achieve technological progress through two approaches: (1) developing and enhancing their domestic technology or (2) acquiring advanced technology through international technology diffusion. Recent studies have demonstrated that, in most countries, the primary sources of the technologies contributing to economic development are gained from international technology diffusion rather than developed domestically (e.g., Glass & Saggi 1998) [18]. Prior economics and technology policy research has studied international technology transfer through diverse channels such as trade, foreign

direct investment (FDI), global networks, and joint ventures (Keller & Chinta, 1990 [19]; Reddy & Zhao, 1990 [20]; Cusumano & Elenkov, 1994 [21]).

Intellectual Property Rights (IPRs) have also been considered one of the critical factors of technology transfer (Branstetter et al., 2006) [7]. IP management provides companies with competitive advantages over rivals in the market by encouraging innovation and technological development. A growing body of literature (e.g., Pitkethly, 2001 [22]; Dedrick et al., 2013 [15]; Wen et al., 2016 [5]) has examined the strategic use of intellectual property at diverse levels (e.g., organizational, regional, and national levels). For example, at the firm level, in three broad ways, intellectual property management helps firms achieve a better performance: (1) establish a proprietary market, (2) protect their core technologies and business models, and (3) boost research and development and branding effectiveness (Rivera & Kline, 2000) [23].

Countries source technologies from each other through global IP trade. To effectively obtain and utilize IPs, countries should form alliances with the primary holders of IPs and occupy a core position on the global network, because most of the world's R&D and IPs are concentrated in a few developed countries (OECD, 2015) [24]. The direct influence model suggests that technology diffusion is affected by a country's direct neighbors (e.g., Coleman et al., 1966) [25]. Prior IS literature has found that peers influence technology adoption (e.g., social network apps, online content generation, and the home computer). At the firm level, Ge et al. (2023) [26] found that agglomerations of high-tech industries enhance technology transfer across regional industries. For that reason, many countries and firms recognize the benefits of participating in IP collaborations, because new technologies that can bring a superior return on their investment can be invented by working together and using their strengths (e.g., Lu et al., 2024) [27]. By doing so, some countries build a better IP capability through a direct relationship with the primary holders of IPs, thereby licensing or selling their IPs to other countries (e.g., De Rassenfosse, 2012 [28]; Ernst, 2003 [29]). In this way, global networks of financial investment and trade partnerships among nations facilitate countries' IP strategies, creating and promoting new knowledge and technologies. Thus, countries' IP strategies can be determined by their position or embeddedness in the global network. Therefore, we propose the following:

**H1.** *Countries with a direct relationship with other countries in the global finance network are likely to obtain better IP capabilities, thereby exporting more IPs to other countries than they purchase from others.*

On the other hand, there are indirect influences as well as direct relationships between the nations in the global network. Existing research on IS has studied the importance of its indirect influence on a network and identified that its direct effect was not the only factor driving technology and knowledge diffusion (e.g., Zhang et al., 2018 [30]; Marolt et al., 2022 [31]). Some studies have even found that, in a particular situation, the impact of an indirect influence in a network was more substantial than that of a direct influence on innovation diffusion (e.g., Burt, 1987 [32]; Strang & Tuma, 1993 [33]). However, the results of those prior studies do not support the role of indirect influences in creating and discovering new knowledge; rather, they highlight the ability of indirect influences to adopt innovations (i.e., technology and knowledge) and disseminate them across the network. Countries with indirect relationships in the global network do not have strong IP capabilities to develop but are good at recognizing and importing the IPs they need. As a result, they are prone to purchasing IPs from a group of developed countries to acquire new technologies or speed up their R&D efforts rather than creating them through domestic inventions or collaborating with core IP holders. Thus, we propose the following hypothesis for testing:

**H2.** *Countries with indirect relationships with other countries in the global finance network are likely to import more IPs from others than those export to them.*

From the external perspective, financial investment connectedness determines the cash flow among countries and embeds implicit knowledge in the network. In addition, from the internal perspective, a country's ICT adoption level significantly determines its citizens' access to information. ICT usage is proven to affect countries' economic productivity (e.g., Dedrick et al., 2013 [15]), political corruption (e.g., Srivastava et al. [16], 2016), and social well-being (e.g., Ganju et al., 2016 [17]). With the advancement of information technology, international collaboration has become more effective and efficient (Zhang & Dawes, 2006) [34]. ICT works as an enabler of innovation to create new products and services. In addition, the adoption of IT by governments helps countries foster their ability to connect and collaborate with other countries and firms across the globe. Thus, because of the impact of ICT, we assume that countries with high-level ICT usage and infrastructure do not need to depend heavily on the global investment network to acquire new technologies and knowledge. Therefore, we hypothesize the following:

**H3.** *ICT usage mitigates the impact of global financial investment connectedness on countries' IP strategies.*

### 3. Research Methodology

The unit of analysis used in this study is a country. Multiple data sources were employed. International financial investment data were gathered from the Coordinated Portfolio Investment Survey (CPIS) held in the International Monetary Fund (IMF) library (http://www.elibrary.imf.org/, accessed on 5 March 2024). The data on ICT, intellectual property, and other control variables were collected from the World Bank database (http://data.worldbank.org/, accessed on 5 March 2024). Finally, we collected data on 264 countries from 2001 to 2015. Our final number of data observations was 3960.

We measure a country's IP strategy using two dimensions: payments and receipts for IP use. Figure 1a shows that some countries have more earnings from IP than payments for using other countries' IP. Other countries purchase more IPs from other countries instead of selling them to others. Thus, a country's ratio of IP receipts to IP payments indicates how much they rely on themselves to develop products or services. Therefore, we have created a dummy variable, IP strategy, to demonstrate whether a country receives more charges than payments for IPs. If the ratio is greater than 1, we label the country's IP strategy as a proactive strategy (i.e., they have a trade surplus in IP), for which its receipt is higher than its payment for IP usage. On the contrary, a country's IP strategy is denoted as a dependence strategy (i.e., they have a trade deficit in IP) if its receipt is lower than its payment for IP. The distribution of these coded IP strategies is shown in Figure 1b.

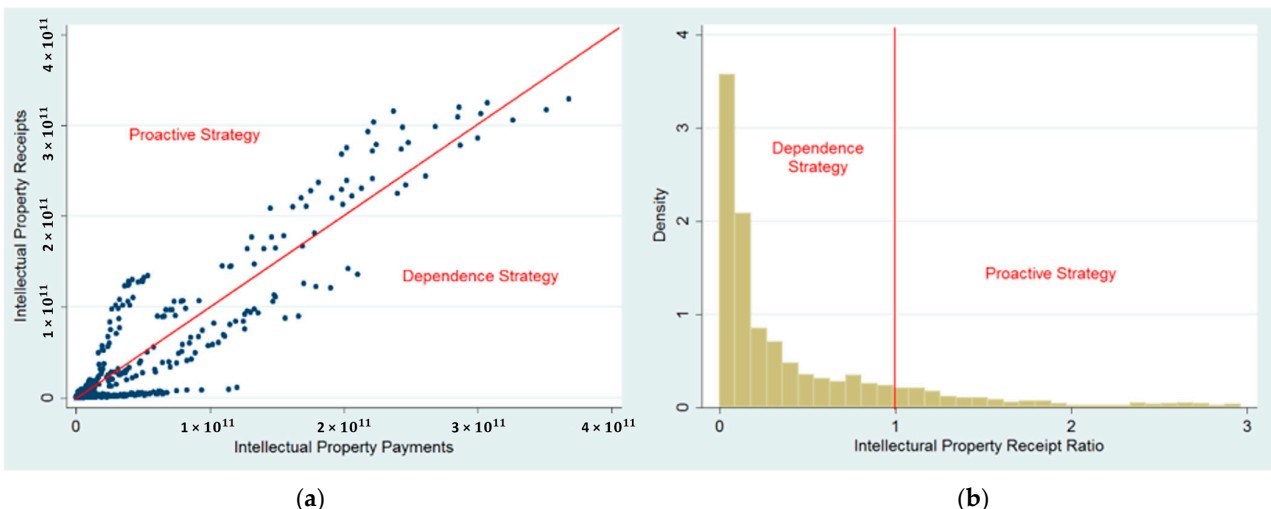

(**a**)

(**b**)

**Figure 1.** Measurement of intellectual property strategies. (**a**) Dimensions of IP Strategy; (**b**) Histogram of IP Receipt Ratio.

To measure global financial investment connectedness, we used a social network analysis (SNA). SNA have been widely used in social science research, along with the growing availability of and access to user data on social media platforms through free APIs (Moreno-Sandoval [35]). We constructed a directed and unweighted network based on the CPIS data. Each country in the network represents a specific node. A connection happens when one country invests in another country. The network was constructed on a yearly basis. Every year, the number of investments between countries has increased. For example, in 2001, global financial investment was 12,676; it rose to 17,314 in 2005, 23,080 in 2010, and 357,731 in 2015.

We calculated the degree of centrality to quantify a country's connectedness within the global financial network. Degree centrality refers to the total number of nodes directly connected to the focal node. Since the global investment network is direct, two variables, Indegree and Outdegree, were used to measure the degree of centrality of incoming connectedness and outgoing connectedness, respectively. Along with the increasing investment between countries, the investment network is becoming more and more complicated. In addition, more central nodes (i.e., represented as red circles) are appearing, and the sizes of these central nodes are increasing. However, degree centrality only captures the direct connectedness of countries in this network. There are indirect relationships among nations in the global network. Thus, we consider the variable of betweenness to represent the betweenness of a country in the global financial investment network. Betweenness centrality not only considers the nodes directly connected to the focal node but also measures its indirect connections in the network.

In line with prior research (Ganju et al., 2016) [17], ICT was indicated by three major variables: fixed telephone lines, internet users, and mobile users out of 100 people. The variable *ICT* was thus created to represent the total number of users across these three categories. In addition, we also control for the confounding factors that may affect a country's IP strategy, including the *GDP*, *Population*, *Education*, and *FDI* inflow, by referring to prior academic studies and industry reports (e.g., Gilbert, 1990 [36]) (OECD Report— https://www.oecd.org/sdd/leading-indicators/49985449.pdf, accessed on 5 March 2024). The net rate of secondary school attendance was measured as *Education*. All the variables were transformed by logarithm.

Considering the panel data's structure and the attributes of the dependent variable, we utilized the panel logit model to estimate the impacts of global financial investment connectedness and ICT on IP strategy choice. Our data from multiple sources contain a large amount of missing data, so many subjects were not measured across the same number of time points. Since the random effect model has a particular advantage in analyzing longitudinal data with nonignorable missing data on the measured subjects (e.g., Hedeker & Gibbons, 1997 [37]; Shah et al., 1997 [38]), we used the random effect model. Prior studies have used the random effect model for analyzing data that have been missing for a long time (e.g., Laird & Ware, 1982) [39]. We simultaneously lagged all the independent and control variables to avoid potential endogeneity issues. Furthermore, since there might be some unobserved time-related factors, such as the technological cycle, we included a time dummy to control this unobserved heterogeneity. The estimation model is given as:

$$
\begin{aligned}
\Pr(\text{IP Strategy}_{i,t+1} = 1) = {} & \beta_0 + \beta_1 \text{Indegree}_{it} \\
& + \beta_2 \text{Outdegree}_{it} + \beta_3 \text{Betweenness}_{it} + \beta_4 \text{ICT}_{it} \\
& + \beta_5 \text{Indegree}_{it} \times \text{ICT}_{it} + \beta_6 \text{Outdegree}_{it} \times \text{ICT}_{it} \\
& + \beta_7 \text{Betweenness}_{it} \times \text{ICT}_{it} + \beta_8 \text{GDP}_{it} + \beta_9 \text{FDI}_{it} \\
& + \beta_{10} \text{Education}_{it} + \beta_{11} \text{Population}_{it} + \alpha_t + \varepsilon_{it}
\end{aligned}
$$

where $\alpha_t$ represents the time-specific dummy and $\varepsilon_{it}$ refers to the error term.

## 4. Results

Table 1 shows the descriptive statistics and correlation of all variables used in our regression analysis. On average, around 54% of countries in our sample adopt a proactive strategy. Furthermore, the correlation results indicate that there is no severe multicollinearity problem. The results of the regression analysis are summarized in Table 2. The first column shows the regression results of the testing of our main hypothesis. In addition, we also utilized the probit model to show the robustness of these results. In addition, we also split the dependent variable by means of conducting additional robustness checks. The results of this, as shown in the other three columns, demonstrate the robustness of our results. We thus report our results based on the testing of our main hypothesis (i.e., the first column).

**Table 1.** Descriptive statistics and correlations.

| | Mean | Std. Dev | 1 | 2 | 3 | 4 | 5 | 6 | 7 | 8 |
|---|---|---|---|---|---|---|---|---|---|---|
| 1. IP Strategy$_{i,t+1}$ | 0.54 | 0.50 | | | | | | | | |
| 2. Indegree$_{it}$ | 0.88 | 0.96 | −0.11 | | | | | | | |
| 3. Outdegree$_{it}$ | 0.47 | 0.97 | −0.04 | 0.42 | | | | | | |
| 4. Betweenness$_{it}$ | 0.78 | 1.86 | −0.07 | 0.52 | 0.59 | | | | | |
| 5. ICT$_{it}$ | 4.22 | 1.29 | −0.17 | 0.46 | 0.20 | 0.24 | | | | |
| 6. GDP$_{it}$ | 8.34 | 1.56 | −0.05 | 0.53 | 0.21 | 0.26 | 0.60 | | | |
| 7. Education$_{it}$ | −0.05 | 0.17 | −0.07 | 0.28 | 0.10 | 0.14 | 0.54 | 0.50 | | |
| 8. Population$_{it}$ | 15.99 | 3.04 | −0.42 | −0.05 | −0.10 | −0.07 | −0.08 | −0.19 | −0.19 | |
| 9. FDI$_{it}$ | 21.22 | 3.12 | −0.39 | 0.19 | 0.02 | 0.07 | 0.44 | 0.45 | 0.11 | 0.73 |

**Table 2.** Investment connectedness, ICT, and IP strategies.

| Variables | Hypothesis Testing | Robustness Checks | | |
|---|---|---|---|---|
| | | Probit Model | Spilt the DV by the Mean | |
| | Logit Model | | Logit Model | Probit Model |
| Indegree | 2.820 ** (1.154) | 1.510 ** (0.642) | 2.135 * (1.232) | 1.209 * (0.704) |
| Outdegree | 0.955 (1.251) | 0.565 (0.746) | 1.509 (1.208) | 0.884 (0.735) |
| Betweenness | −1.785 ** (0.815) | −0.974 ** (0.458) | −1.754 * (0.909) | −0.974 ** (0.492) |
| ICT | 0.212 (0.311) | 0.122 (0.172) | 0.179 (0.395) | 0.094 (0.218) |
| ICT × Indegree | −0.707 *** (0.247) | −0.376 *** (0.139) | −0.642 ** (0.285) | −0.358 ** (0.162) |
| ICT × Outdegree | −0.190 (0.280) | −0.112 (0.167) | −0.300 (0.270) | −0.176 (0.162) |
| ICT × Betweenness | 0.399 ** (0.185) | 0.215 ** (0.104) | 0.417 ** (0.209) | 0.229 ** (0.112) |
| GDP | −0.579 * (0.339) | −0.327 (0.215) | −1.450 *** (0.457) | −0.810 *** (0.282) |

**Table 2.** *Cont.*

| Variables | Hypothesis Testing | Robustness Checks | | |
|---|---|---|---|---|
| | | Probit Model | Spilt the DV by the Mean | |
| | Logit Model | | Logit Model | Probit Model |
| FDI | 0.112 (0.109) | 0.064 (0.062) | 0.125 (0.132) | 0.071 (0.074) |
| Education | 0.423 (1.750) | 0.179 (1.025) | 2.106 (1.906) | 1.137 (1.128) |
| Population | −1.137 *** (0.179) | −0.642 *** (0.119) | −1.840 *** (0.246) | −1.036 *** (0.164) |
| Time Fixed Effect | Yes | Yes | Yes | Yes |
| Wald $\chi^2$ | 122.47 *** | 127.36 *** | 143.75 *** | 150.48 *** |

Note—number of observations: 2326; number of countries: 222; robust standard errors are in parentheses; significance levels: *** $p < 0.01$, ** $p < 0.05$, * $p < 0.1$.

First, incoming degree centrality positively affects a country's possibility of choosing a proactive IP strategy. If a country attracts more countries' financial investments, it is more likely to initiate a proactive IP strategy by selling more IPs than it purchases from others. Therefore, we confirm that H1 is supported. However, the betweenness centrality negatively and significantly influences a country's likelihood of implementing a proactive IP strategy. Suppose a country has more indirect connections in the global financial investment network. In that case, it is more likely to initiate a dependence IP strategy, buying IPs abroad rather than developing them domestically. Thus, H2 is supported. Surprisingly, a country's outgoing connections in the global financial investment network will not significantly affect its choice of IP strategy.

In addition, the positive impact of incoming degree centrality on a country's choice of a proactive IP strategy is decreased by its level of ICT usage. However, the level of ICT usage increases the negative effect of betweenness centrality on a country's choice of a proactive IP strategy. Overall, we can observe that ICT weakens an IP strategy's reliance on global financial investment connectedness. Therefore, H3 is supported. In terms of the control variables, a country's population and GDP negatively affect its likelihood of initiating a proactive IP strategy.

## 5. Conclusions

This paper investigated the direct and indirect impacts of global financial investment connectedness and ICT on countries' IP strategies. By utilizing the panel logit model on longitudinal country-level data from 2001 to 2015, we found that countries' positions in the global financial investment network significantly affect their IP strategies. In particular, if a country attracts a larger number of financial investments from other countries, it is more likely to initiate a proactive IP strategy by selling more IPs than it purchases from others. However, if a country has more indirect connections in the global financial investment network, it is more likely to initiate a dependence IP strategy. Furthermore, ICT usage weakens the IP strategy's reliance on global financial investment connectedness. Specifically, the positive impact of incoming degree centrality on a country's choice of a proactive IP strategy is decreased by its level of ICT usage. However, the level of ICT usage increases the negative effect of betweenness centrality on a country's choice of a proactive IP strategy. We suggest that governments align their global investment portfolio and ICT usage to improve IP strategies.

This study offers several important theoretical and practical implications for IP management research and the literature on innovation and technology diffusion. First, it theoretically contributes to IP management research. Whereas the prior literature on IP management has focused on the impact of IP development and found mixed results

about the impacts of IP protection and that IP protection promotes technology diffusion (e.g., Germeraad, 1999 [40]; Ernst, 2003 [11]), this study investigates a country's strategic behaviors in terms of exporting their IPs or importing IPs from other countries. From the new insight provided by this study, we learn how a country determines its IP strategy and how important its strategic position in the global network is in shaping its IP strategy. Future studies should follow on from this to discover other the behavioral, financial, and technical aspects of countries that affect their IP strategies.

In addition, this study contributes to the innovation diffusion literature. So far, the prior literature on the diffusion of innovation has focused on five main elements (i.e., innovation itself, communication channels, adopters, time, and social systems) and identified that an innovation must be widely adopted so that it reaches critical mass (Rogers, 2010) [41]. These earlier studies mainly introduced the characteristics (e.g., ability, motivation, compatibility, observability, etc.) of individual adopters and organizations, which were good predictors of technology adoption. Unlike prior research, this study has explored nations' export or import IP rights, which are considered the consequences of their innovations, and not their individual and organizational levels. Also, by highlighting the importance of a country's position in the global network, this study discovers how connectedness and its position in the network influence a country's IP strategies, thereby stimulating the diffusion of innovation across the globe. Along with the adopters' identified individual or organizational characteristics, the surrounding environments of these entities can influence and lead them to choose different IP strategies.

This research contributes to the literature on the value and effectiveness of information systems by identifying that a country's ICT usage lessens the influence of its network position on its IP strategies. The study results show that a country's network position and connectedness to the global financial network influence its IP strategy. In addition, a country's ICT usage mitigates its dependence on the global financial investment network. It demonstrates that ICT usage enables governments to develop new knowledge and technologies or collaborate with other countries to promote innovation. Prior studies on the impact of ICT also have indicated that ICT usage affects diverse aspects of countries, including their economic productivity, $CO_2$ emissions, political corruption, and social well-being (e.g., Danish et al., 2018 [42]). South Korea, for example, demonstrates a remarkably high level of ICT utilization. South Korea's ICT industry has grown steadily over the past few decades, producing globally recognized companies.

A notable example is Samsung Electronics, which develops various ICT products and technologies and holds patents and exports them. Through the development of such companies, South Korea not only relies on intellectual property imports but also exports its own patents and technologies to the international market, securing income from intellectual property. This study contributes to the literature by uncovering ICT as an alternative channel for transferring new technologies and other human intellect to the global network, supplementing traditional IP trade.

These research findings indicate that using ICT reduces a nation's external dependence on intellectual property and offers several implications from a sustainability perspective. First, utilizing ICT to develop and protect domestic intellectual property obviates the need to import technology or intellectual property from abroad, reducing resource consumption and mitigating adverse environmental impacts. Research and development efforts can prioritize minimizing energy consumption and resource usage when developing new technologies.

In addition, leveraging ICT to develop and protect domestic intellectual property can facilitate technology transfer and knowledge sharing. This fosters a greater dissemination and propagation of knowledge, promoting technological innovation and advancement. It also helps reduce knowledge disparities and enhances the cooperation between local communities and nations.

Third, nations can enhance their economic autonomy by reducing their external dependence by developing and protecting domestically generated technology and intellectual

property. A robust indigenous technological capability and knowledge economy bolster a country's economic stability in the long term, supporting sustainable national development. Last, utilizing ICT to develop and protect domestic intellectual property provides opportunities for different social groups, promoting social inclusivity and equality. Diverse knowledge and technological expertise across various industries foster innovation and creativity, contributing to societal advancement.

In conclusion, ICT can help many developing countries save energy and resources (e.g., financial and human resources) instead of overspending on building and joining the global knowledge alliance. These implications provide a conceptual framework for understanding how safeguarding a nation's intellectual property through ICT can contribute to sustainable development. Integrating these concepts into actual policy and strategy formulations is crucial for advancing both intellectual property protection and sustainable development goals.

This study also has some practical implications for governments around the world. First, from the results of this study, we learn that government officials should decide on their IP strategy based on their strategic position in the global financial network. Key stakeholders, such as their main trade partners in the global network, should vary depending on the countries' IP strategies. In addition, a country's ICT capabilities can mitigate its dependency on the global network, thereby allowing it to adjust its IP strategy. Governments should understand the importance of investing in ICT in terms of IP development.

We next note some limitations and discuss opportunities for future research. First, this study uses countries' financial investments as the research context to explore their IP strategies. Future research could try to test its external validity by replicating our results in other contexts, such as countries in other types of networks or other contexts. In addition, this study focuses on the relationship between IP strategies and countries' positions in the global financial network. Future research could focus on firms' connectedness or position in financial investment networks and their IP strategy behaviors.

**Author Contributions:** Conceptualization, Y.K.; methodology, Y.K.; software, Y.K.; validation, S.Y. and Y.K.; formal analysis, S.Y.; investigation, S.Y.; resources, S.Y.; data curation, Y.K.; writing—original draft preparation, S.Y.; writing—review and editing, S.Y.; visualization, Y.K.; supervision, S.Y.; project administration, S.Y.; funding acquisition, S.Y. All authors have read and agreed to the published version of the manuscript.

**Funding:** This work was supported by the Soongsil University Research Fund (New Professor Support Research) of 2022 (No. 202210001439).

**Institutional Review Board Statement:** Not applicable.

**Informed Consent Statement:** Not applicable.

**Data Availability Statement:** These data were derived from the following resources available in the public domain: [http://www.elibrary.imf.org/; http://data.worldbank.org/, accessed on 5 March 2024].

**Conflicts of Interest:** Author Yonghee Kim was employed by the company OpenRoute. The remaining authors declare that the research was conducted in the absence of any commercial or financial relationships that could be construed as a potential conflict of interest.

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
