# Peer review of "Global Financial Investment Connectedness, ICT, and Intellectual Property Strategies: A Country-Level Empirical Analysis"

_sustainability, doi:10.3390/su16083282_

Round 1

Reviewer 1 Report

Comments and Suggestions for Authors

This is a highly interesting and well-founded contribution. I have just few suggestions for its improvement:

- please explain your sample - which countries (jurisdictions) are invovled and why;

- please explain more your charts, tables and graphs and perhaps make maps in figure 5 bigger, so we can see these networks in more detail;

- please provide examples in your results/discussions part and generally expand it;

- please consider to include more current references and address new studies about similar topics in a comparative manner.

Thanks again for this contribution, it is almost ready to be published!

Comments on the Quality of English Language

Only minor issues, i.e. generally OK.

Reviewer 2 Report

Comments and Suggestions for Authors

 This paper uses country-level data from 2001-2015 to investigate the impact of national financial investment connectedness on intellectual property strategy, while also considering the role of ICT usage. Overall, this article presents an interesting topic, and the research result has potential contributions and application value. However, in some aspects of this article, the authors may consider making the following improvements.

1. Many important concepts in this article are not clearly defined, including but not limited to: global finance network, IP strategy, IP capability, and financial investment connectedness. The authors should provide a clear definition of these terms, as well as indicate the relevant literature sources.

2. This article requires descriptive statistical analysis of the data samples provided. The authors should state how many data points are included in the collected country-level data samples from 2001-2015, and briefly describe the data cleaning process.

3. Figure 2 is too vague, and it is recommended that this article use tables to illustrate the evolution of various network attribute indicators.

4. The theoretical contributions and policy implications of this article are not clear enough and need further improvement.
